# Expression of TMPRSS2 is up-regulated by bacterial flagellin, LPS, and Pam3Cys in human airway cells

Marie Schwerdtner[1],*, Annika Skalik[1],*, Hannah Limburg[1], Jeff Bierwagen[2] , Anna Lena Jung[2], Jens Dorna[3], Andreas Kaufmann[3], Stefan Bauer[3] , Bernd Schmeck[2,4], Eva Böttcher-Friebertshäuser[1] 

**Many viruses require proteolytic activation of their envelope proteins for infectivity, and relevant host proteases provide promising drug targets. The transmembrane serine protease 2 (TMPRSS2) has been identified as a major activating protease of influenza A virus (IAV) and various coronaviruses (CoV). Increased TMPRSS2 expression has been associated with a higher risk of severe influenza infection and enhanced susceptibility to SARS-CoV-2. Here, we found that *Legionella pneumophila* stimulates the increased expression of TMPRSS2-mRNA in Calu-3 human airway cells. We identified flagellin as the dominant structural component inducing TMPRSS2 expression. The flagellin-induced increase was not observed at this magnitude for other virus-activating host proteases. TMPRSS2-mRNA expression was also significantly increased by LPS, Pam3Cys, and *Streptococcus pneumoniae*, although less pronounced. Multicycle replication of H1N1pdm and H3N2 IAV but not SARS-CoV-2 and SARS-CoV was enhanced by flagellin treatment. Our data suggest that bacteria, particularly flagellated bacteria, up-regulate the expression of TMPRSS2 in human airway cells and, thereby, may support enhanced activation and replication of IAV upon co-infections. In addition, our data indicate a physiological role of TMPRSS2 in antimicrobial host response.**

## Introduction

Influenza viruses and coronaviruses (CoV) are major respiratory pathogens with pandemic potential. Influenza A viruses (IAV) and influenza B viruses cause acute respiratory disease (flu) that results in 3–5 million cases of severe respiratory illness and 290,000–650,000 deaths annually (WHO, November 2018). Influenza viruses are enveloped viruses with a segmented, negative-sense, single-stranded RNA genome and belong to the family *Orthomyxoviridae*. IAV circulate in a wide range of animal species with wild aquatic birds being the natural reservoir. The recurrent transmission of avian IAV to other host species provides the basis for the emergence of new viruses posing unpredictable global public health threats and may even provoke an influenza pandemic. The most devastating influenza pandemic in recent history, the 1918 H1N1 "Spanish flu," caused 50 million deaths worldwide. CoV are a large family of single-stranded, positive-sense RNA viruses belonging to the order *Nidovirales*. CoV also infect a broad range of mammalian and avian species, causing respiratory or enteric diseases. Infection of humans with globally circulating human coronaviruses (HCoV) is mainly associated with the common cold. However, zoonotic transmission of animal CoV to humans can result in severe disease as exemplified by the current coronavirus disease 2019 (COVID-19) pandemic caused by severe acute respiratory syndrome–CoV-2 (SARS-CoV-2), as well as the SARS-CoV outbreak in 2002/2003 and the recurrent transmission of Middle East respiratory syndrome (MERS)–CoV to humans.

Infection of both influenza viruses and CoV is initiated by their major surface glycoproteins HA and spike protein (S), respectively, through attachment to cell surface receptors and fusion of the viral lipid envelope and cellular membranes to release the virus genome into the host cell. Like fusion proteins of many other viruses, HA and S are synthesized as fusion-incompetent precursor proteins and require cleavage by host cell proteases to gain their fusion capacity (reviewed in Böttcher-Friebertshäuser [2018] and Hoffmann et al [2018]). Proteolytic activation is essential for virus infectivity and spread, and relevant host proteases provide potential drug targets.

The amino acid sequence at the cleavage site determines which host protease is able to cleave: multibasic motifs of the consensus sequence R-X-R/K-R↓ are cleaved by subtilisin-like proprotein convertases (PC) such as furin and PC5/6, whereas cleavage at a single arginine residue designated as "monobasic cleavage site" is performed by trypsin-like proteases (Klenk et al, 1975; Stieneke-Gröber et al, 1992). We first identified the trypsin-like transmembrane serine protease 2 (TMPRSS2) as a virus-activating protease,

[1]Institute of Virology, Philipps-University Marburg, Marburg, Germany    [2]Institute for Lung Research, Universities of Giessen and Marburg Lung Center, Philipps-University Marburg, German Center for Lung Research (DZL), Marburg, Germany    [3]Institute of Immunology, Philipps-University Marburg, Marburg, Germany    [4]Department of Pulmonary and Critical Care Medicine, Philipps-University Marburg, Marburg, Germany, Member of the German Center for Infectious Disease Research (DZIF), Marburg, Germany

Correspondence: friebertshaeuser@staff.uni-marburg.de
*Marie Schwerdtner and Annika Skalik contributed equally to this work

by demonstrating that it cleaves the HA of human IAV with a monobasic cleavage site (Böttcher et al, 2006). Subsequently, TMPRSS2 was shown to activate the fusion proteins of a number of other respiratory viruses including human metapneumovirus, human parainfluenza viruses, and CoV including SARS-CoV, MERS-CoV, and more recently SARS-CoV-2 in vitro (reviewed in Böttcher-Friebertshäuser [2018]; Bestle et al [2020]; Hoffmann et al [2020]). Remarkably, lack of TMPRSS2 expression prevents multicycle replication and pathogenesis of IAV, and SARS-CoV, MERS-CoV, and SARS-CoV-2 in mice because of inhibition of activation of progeny virus and consequently inhibition of virus spread along the respiratory tract (Hatesuer et al, 2013; Sakai et al, 2014; Tarnow et al, 2014; Iwata-Yoshikawa et al, 2019; Lambertz et al, 2020; Li et al, 2021). Vice versa, single-nucleotide polymorphisms in the *TMPRSS2* gene predicted to cause higher TMPRSS2 expression have been associated as a risk factor for severe IAV infections and more recently with an increased risk of SARS-CoV-2 infection in clinical studies (Cheng et al, 2015; Schönfelder et al, 2021; Pandey et al, 2022).

Co-infecting bacteria are commonly identified in influenza patients and are a major cause of morbidity and mortality. During the 1918 Spanish flu, an estimated 95% of severe infections and deaths occurred concurrently with bacterial pneumonia (McCullers, 2006; Morens et al, 2008; Metersky et al, 2012; Morris et al, 2017; MacIntyre et al, 2018). Despite the introduction of antibiotics and influenza vaccines since then, death because of bacterial pneumonia upon co-infection remains a significant problem. An estimated 30% of influenza-associated deaths during the 2009 H1N1 pandemic and ~44–57% of hospitalizations and 25% of influenza-associated deaths during seasonal outbreaks show co-infections with bacteria (Peltola et al, 2005; Smith & McCullers, 2014). The most common bacteria identified in influenza pneumonia are *Streptococcus pneumoniae* and *Staphylococcus aureus*. IAV has been demonstrated to increase the susceptibility to secondary bacterial infections by multiple factors described in a number of excellent reviews (Bosch et al, 2013; McCullers, 2014; Smith & McCullers, 2014). Interestingly, in 1987 Tashiro et al demonstrated that bacterial infection may also be beneficial for the virus by providing proteases that support HA activation (Tashiro et al, 1987). Some strains of *S. aureus* were shown to secrete proteases that are capable of cleaving the HA of certain influenza strains in vitro. Experimental co-infection of mice resulted in fatal disease with extended lesions in the lungs, whereas infection with either the virus or the bacterium alone did not cause significant symptoms of disease.

In a clinical study, Rizzo et al reported that during the 2009 H1N1 IAV pandemic in Italy, 18% of pneumonia cases were associated with co-infection of pandemic H1N1 IAV (H1N1pdm) and *Legionella pneumophila*, suggesting that *L. pneumophila* may be involved in influenza–bacterial pneumonia more often than estimated (Rizzo et al, 2010). *L. pneumophila* is a facultative, intracellular, Gram-negative bacterium with a single, polar flagellum. It is ubiquitously found in freshwater and known to replicate in amoebae, but is also capable of infection and replication within human alveolar macrophages and infection of respiratory epithelial cells (Winn & Myerowitz, 1981; Kwaik, 1998). *L. pneumophila* is the causative agent of Legionnaires' disease, a life-threatening pneumonia, and Pontiac fever, a self-limiting flu-like syndrome. Transmission to humans occurs through contaminated aerosols generated from

man-made water sources including cooling towers, showers, humidifiers, and whirlpools.

Here, we show that *L. pneumophila* stimulates increased TMPRSS2-mRNA expression in Calu-3 human airway cells and that this is primarily due to bacterial flagellin. In addition, LPS and the synthetic lipopeptide Pam3Cys induced a significant increase in TMPRSS2-mRNA expression in Calu-3 cells. We furthermore investigated the expression of further virus-activating proteases upon flagellin stimulation in Calu-3 cells and examined multicycle replication of IAV, SARS-CoV, and SARS-CoV-2 in flagellin-treated Calu-3 cells and primary human bronchial epithelial cells (HBEC) in this study.

# Results

### *L. pneumophila* stimulates the increased expression of TMPRSS2 in Calu-3 human airway cells

In the present study, we aimed to investigate whether *L. pneumophila* may alter the expression of virus-activating proteases in human airway cells. Therefore, we used the human airway epithelial cell line Calu-3 that has been demonstrated to provide a suitable model to study proteolytic activation of IAV and other respiratory viruses in human airway cells (Laporte et al, 2019; Limburg et al, 2019; Bestle et al, 2020). Calu-3 monolayers were stimulated with either viable or heat-inactivated *L. pneumophila* strain Corby wild type (WT) for 24 h. Cells without *L. pneumophila* stimulation served as a control. At 24 h post-infection, total RNA was isolated and the mRNA expression of TMPRSS2 and the related protease TMPRSS4 was determined by RT–qPCR. Induction of IL-6 mRNA expression was analyzed as a control. As shown in Fig 1A, IL-6 mRNA expression was significantly increased in cells inoculated with viable *L. pneumophila* WT or stimulated with heat-inactivated bacteria. Interestingly, TMPRSS2-mRNA expression was increased sevenfold to 10-fold in Calu-3 cells stimulated with either viable or inactivated bacteria. In contrast, no marked up-regulation of TMPRSS4-mRNA expression by *L. pneumophila* was observed in the cells. The data revealed that *L. pneumophila* increases the expression of TMPRSS2-mRNA but not of TMPRSS4-mRNA in Calu-3 cells independent of viability, suggesting that a structural component stimulates TMPRSS2-mRNA expression.

### Flagellin is the major component of *L. pneumophila* that induces TMPRSS2-mRNA expression in Calu-3 cells

Next, we asked which structural component of *L. pneumophila* affects TMPRSS2-mRNA expression in Calu-3 cells. *L. pneumophila* is a flagellated, Gram-negative bacterium. Bacterial flagella are helical proteinaceous fibers, composed of the protein flagellin, that confer motility to many bacteria. TLR5 recognizes an evolutionarily conserved site on bacterial flagellin (Smith et al, 2003). To examine whether *L. pneumophila* flagellin stimulates TMPRSS2-mRNA expression in Calu-3 cells, we incubated cells with viable *L. pneumophila* WT or a flagellin-deficient mutant (Δfla) (Schmeck et al, 2006, 2008) for 24 h, and then, mRNA levels of TMPRSS2, TMPRSS4,

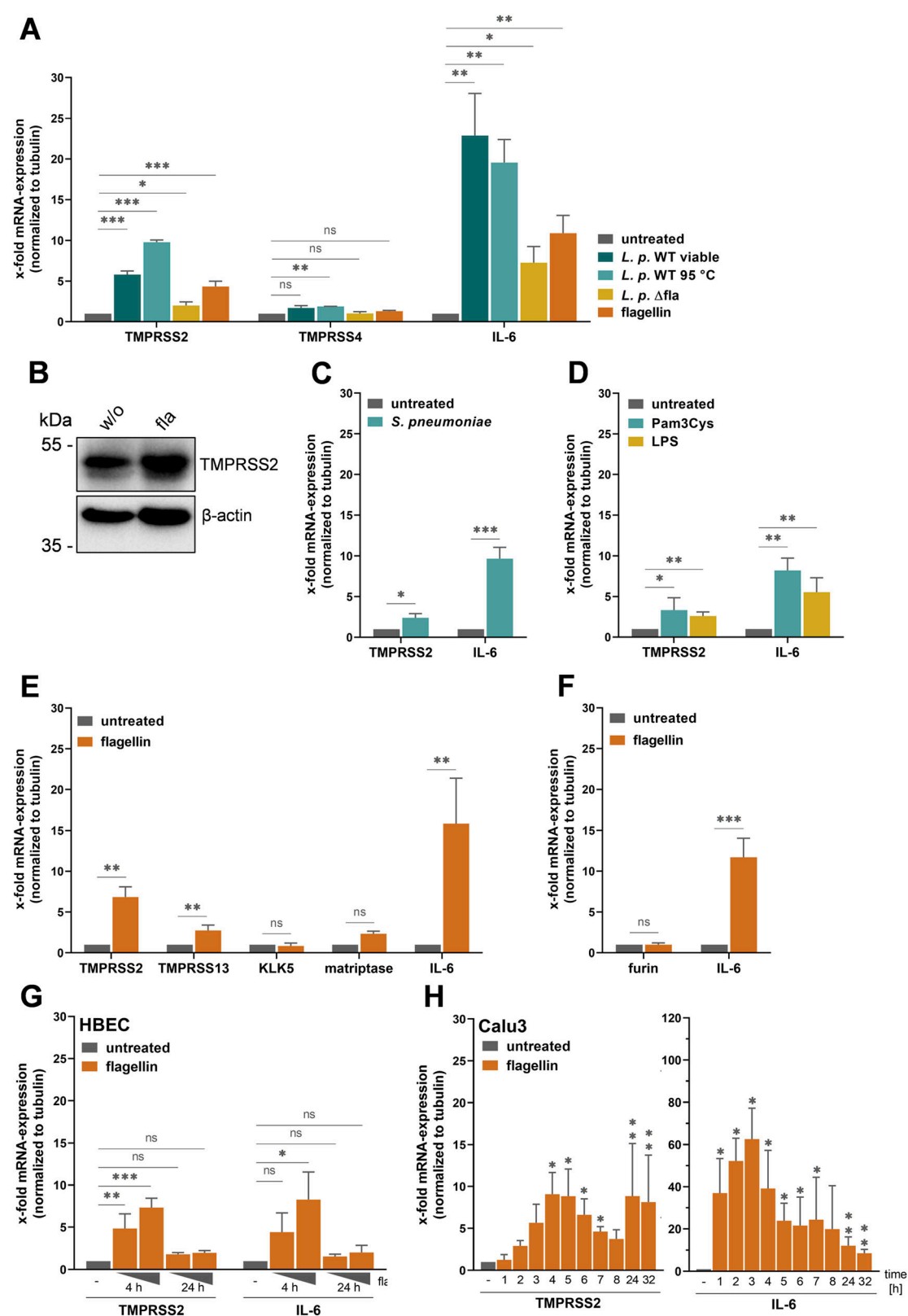

**Figure 1.** *L. pneumophila* **flagellin induces TMPRSS2-mRNA expression in Calu-3 human airway cells.**
**(A)** Cells were inoculated with either viable or heat (95°C)-inactivated *L. pneumophila* (*L.p.*) WT, viable flagellin-deficient mutant (*L.p.* Δfla), or purified flagellin from *Salmonella typhimurium* (200 ng/ml) for 24 h. Untreated cells were used as controls. Total RNA was isolated, and relative mRNA expression levels of TMPRSS2, TMPRSS4, and IL-6 were measured by RT–qPCR and normalized using tubulin as an internal control. Data are mean values + SD of three independent experiments. **(B)** Calu-3 cells

and IL-6 were determined by RT–qPCR (Fig 1A). As described previously, the Δfla mutant induced a lower increase in IL-6 mRNA expression compared with the WT strain (Schmeck et al, 2008). Interestingly, the Δfla mutant increased TMPRSS2-mRNA expression only twofold in Calu-3 cells in contrast to a sixfold increased expression induced by the WT strain, indicating that flagellin is the dominant component of *L. pneumophila* that stimulates TMPRSS2-mRNA expression in the cells. To confirm this, Calu-3 cells were treated with commercially available flagellin purified from *Salmonella enterica* serovar Typhimurium for 24 h. Subsequent RT–qPCR analysis revealed that TMPRSS2-mRNA expression was increased fivefold in flagellin-treated cells compared with control cells (Fig 1A). In contrast, TMPRSS4-mRNA expression was not significantly altered by flagellin. Furthermore, increased TMPRSS2 protein expression was detected in flagellin-treated Calu-3 cells compared with untreated control cells by Western blot analysis (Fig 1B). Together, the data show that flagellin stimulates TMPRSS2 expression in Calu-3 human airway cells.

### Induction of TMPRSS2-mRNA expression by other bacterial components

We then examined whether the non-flagellated, Gram-positive bacterium *S. pneumoniae* that is commonly identified as a co-infecting pathogen in influenza pneumonia is able to stimulate TMPRSS2 expression in Calu-3 cells. Calu-3 monolayers were inoculated with viable *S. pneumoniae* for 24 h. A slight but significant increase in TMPRSS2-mRNA expression was observed in *S. pneumoniae*–stimulated Calu-3 cells, however less pronounced compared with flagellin treatment (Fig 1C). IL-6 mRNA expression was strongly induced by *S. pneumoniae* (Fig 1C). The data suggested that other bacterial components may induce TMPRSS2 expression as well. Therefore, we examined whether LPS (the major outer surface membrane component of Gram-negative bacteria, recognized by TLR4) or lipoproteins of Gram-positive bacteria (recognized by TLR2) are able to induce enhanced TMPRSS2-mRNA expression in Calu-3 cells. Calu-3 monolayers were stimulated with the synthetic TLR1/TLR2 agonist Pam3Cys or with LPS for 24 h; subsequently, RNA was isolated and the expression of TMPRSS2-mRNA was analyzed. As shown in Fig 1D, a twofold to threefold increase in TMPRSS2-mRNA expression was observed in Pam3Cys- or LPS-treated Calu-3 cells. Induction of IL-6 mRNA expression by Pam3Cys or LPS, however, was reduced compared with stimulation of cells by flagellin, indicating that response to these bacterial components is less efficient in Calu-3 cells.

Taken together, the data indicate that bacterial components, particularly flagellin and to a minor extent LPS and lipopeptide, stimulate the increased expression of the virus-activating protease TMPRSS2 in Calu-3 human airway cells.

### Expression of further virus-activating proteases upon flagellin stimulation in Calu-3 cells

Next, we examined whether the expression of further trypsin-like proteases present in Calu-3 cells (Laporte et al, 2019; Limburg et al, 2019) that have been shown to be capable of cleaving IAV HA in vitro, such as TMPRSS13, KLK5, and matriptase, is altered by flagellin stimulation. A slight but significant increase in TMPRSS13-mRNA expression was observed in flagellin-treated Calu-3 cells, however less pronounced compared with up-regulation of TMPRSS2-mRNA expression by flagellin (Fig 1E). In contrast, KLK5- or matriptase-mRNA levels were not significantly altered upon flagellin treatment. Furthermore, we found that the mRNA expression of furin, which activates the envelope proteins of a broad number of viruses at multibasic motifs, was not affected by flagellin stimulation in Calu-3 cells (Fig 1F).

In sum, marked up-regulation of mRNA expression by flagellin was observed primarily for TMPRSS2 and to a minor extent for TMPRSS13 in Calu-3 cells, but not for any of the other virus-activating proteases tested.

### TMPRSS2-mRNA expression is increased by flagellin in primary HBEC

Calu-3 is a cell line obtained from a lung adenocarcinoma tissue. To examine whether TMPRSS2 expression is stimulated by flagellin in primary human respiratory cells as well, we determined the expression of TMPRSS2-mRNA in primary HBEC upon flagellin stimulation. Flagellin treatment of HBEC showed no marked increase in IL-6 mRNA expression at 24 h even at a higher dose of 1 $\mu$g/ml flagellin (Fig 1G). However, an increase in IL-6 mRNA expression was visible after 4 h of flagellin stimulation. TMPRSS2-mRNA expression was significantly increased (4.8- and sevenfold) in cells treated with flagellin for 4 h in a dose-dependent manner. The data show that TMPRSS2 expression is also increased by flagellin in primary human bronchial cells.

Because we saw in HBEC that TMPRSS2-mRNA expression was already increased after 4 h, whereas no significant effect was observed after 24 h, we looked at the time course of TMPRSS2-mRNA expression after flagellin stimulation in Calu-3 cells over 32 h. A first peak in TMPRSS2-mRNA expression could indeed be observed after

---

were stimulated with flagellin (200 ng/ml) or remained untreated (w/o) for 24 h. Cell lysates were subjected to reducing SDS–PAGE and Western blot analysis using a TMPRSS2-specific antibody. β-actin was used as a loading control. **(C)** Calu-3 monolayers were inoculated with *S. pneumoniae* and incubated for 24 h. Total RNA was isolated, and relative mRNA expression levels of TMPRSS2 and IL-6 were measured by RT–qPCR as described above. Data are mean values + SD (n = 3). **(D)** Cells were treated with flagellin (200 ng/ml), LPS (1 $\mu$g/ml), or Pam3Cys (5 $\mu$g/ml) or remained untreated for 24 h. The relative mRNA expression of TMPRSS2 and IL-6 was measured by RT–qPCR. Data are mean values + SD of three independent experiments. **(E, F)** Calu-3 monolayers were incubated with flagellin for 24 h (200 ng/ml) or remained untreated. **(E, F)** Total RNA was isolated, and relative mRNA expression levels of the indicated trypsin-like proteases (E) or furin (F) were analyzed by RT–qPCR. Data are mean values + SD of three independent experiments. **(G)** Primary human bronchial epithelial cells were stimulated with flagellin (200 ng/ml or 1 $\mu$g/ml) for 4 or 24 h. Untreated cells were used as a control. Total RNA was isolated, and the relative mRNA expression of TMPRSS2 and IL-6 was analyzed by qRT–PCR. Data shown are mean values + SD of three experiments. **(H)** Calu-3 cells were treated with 200 ng/ml flagellin for the indicated time. Total RNA was isolated and analyzed by qRT–PCR with TMPRSS2- and IL-6–specific primers, respectively. Data are mean values of three independent experiments. Statistics: a $t$ test with $\Delta C_t$-values, *$P$ < 0.05, **$P$ < 0.01, and ***$P$ < 0.001 compared with the untreated control sample.

4 h of flagellin stimulation in Calu-3 cells (Fig 1H). The expression then gradually decreased. After 24 h of flagellin treatment, increased TMPRSS2-mRNA expression was again observed, which persisted until the 32-h time point. IL-6 mRNA expression was highest after 3 h of flagellin stimulation, but could also be detected at all other time points.

## Inhibition of p38 MAPK, ERK, and NF-κB reduces flagellin-induced TMPRSS2-mRNA expression in Calu-3 cells

Flagellin has been shown to activate p38 MAPK, extracellular signal-regulated kinases (ERK), and NF-κB transcription factors in a TLR5-dependent manner (Hayashi et al, 2001; Yu et al, 2003). To gain further insight into signaling pathways underlying flagellin-induced TMPRSS2 expression in Calu-3 cells, stimulation of the cells with *L. pneumophila* WT was performed in the presence of inhibitors of p38 MAPK (SB202190), ERK (U0126), or NF-κB (BAY 11-7821). Treatment of Calu-3 cells with the inhibitors at a concentration of 10 μM for 24 h displayed no cytotoxicity (data not shown). Blocking of p38 markedly reduced the expression of TMPRSS2-mRNA and IL-6 mRNA in flagellin-stimulated cells (Fig 2A). Inhibition of ERK also reduced TMPRSS2-mRNA expression, but less pronounced compared with p38 inhibition, and did not reduce IL-6 mRNA expression in Calu-3 cells. Inhibition of NF-κB activation strongly reduced TMPRSS2 and IL-6 expression in flagellin-stimulated Calu-3 cells (Fig 2B). Together, the data suggest that flagellin-induced up-regulation of TMPRSS2 expression is dependent on the p38 MAPK signaling pathway and the transcription factor NF-κB, and, to a lesser extent, the ERK pathway.

## Multicycle replication of IAV, SARS-CoV-2, and SARS-CoV in flagellin-treated Calu-3 cells

The data suggested that particularly flagellated bacteria cause the increased expression of TMPRSS2 in human airway cells and, thereby, may support enhanced activation of influenza viruses or CoV upon co-infections. Therefore, we investigated multicycle replication of human pandemic H1N1pdm and H3N2 IAV in flagellin-treated Calu-3 cells. Calu-3 monolayers were pre-treated with flagellin for 24 h and were then inoculated with the H1N1pdm or H3N2 virus at a low MOI for 1 h, the inoculum was removed, and the cells were incubated for 48 h. Analysis of virus replication in untreated Calu-3 cells and in the presence of exogenous trypsin served as a negative and positive control, respectively. At indicated time points, supernatants were collected and virus titers were determined by the plaque assay. As shown in Fig 3A, replication of the H1N1pdm virus was slightly increased (up to 3.4-fold) in the presence of flagellin at 16–48 h post-infection (p.i.). Virus growth of H3N2 was also increased (up to 4.7-fold) in flagellin-treated cells at early time points p.i., but was not affected at 48 h p.i. Trypsin treatment caused a twofold increase in H1N1pdm titers at 24 h and an up to 3.4-fold increase in H3N2 virus titers at 16–24 h p.i., but did not affect final virus titers. In sum, flagellin stimulation caused a slight, albeit not statistically significant, increase in multicycle replication of H1N1pdm and H3N2 IAV at early time points in Calu-3 cells. Analysis of HA cleavage in H1N1pdm-infected cells at 48 h p.i. revealed a slight increase in the cleavage products HA1 and HA2 in

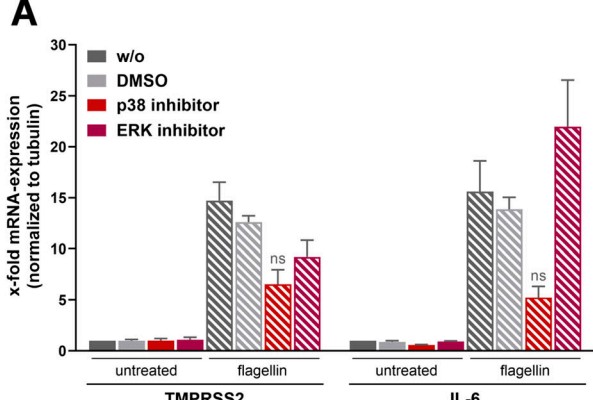

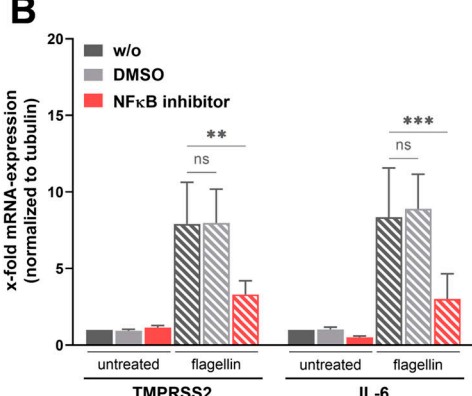

**Figure 2.  p38 MAPK/ERK- and NF-κB–dependent signaling pathways are involved in flagellin-stimulated TMPRSS2 expression.**
**(A, B)** Calu-3 cells were treated with flagellin (200 ng/ml) in the presence (10 μM) or absence of inhibitors of ERK or p38 MAPK (A), or NF-κB (B) for 24 h. Untreated and DMSO-treated cells served as controls. mRNA expression levels of TMPRSS2 and IL-6 were analyzed at 24 h post-treatment by RT–qPCR. Statistics: a *t* test with ΔC$_t$-values, *P < 0.05, **P < 0.01, and ***P < 0.001 compared with the corresponding untreated control sample.

trypsin- or flagellin-treated cells in comparison with control cells (Fig 3B). HA0 was not completely cleaved under any of the conditions, consistent with previous studies on HA cleavage in human cells (Calu-3, primary type II pneumocytes, Caco-2) in the presence and absence of exogenous trypsin (Bertram et al, 2010; Böttcher-Friebertshäuser et al, 2011; Harbig et al, 2020).

In addition, we aimed to investigate whether the increased expression of TMPRSS2 because of flagellin treatment may support enhanced susceptibility and replication of SARS-CoV and SARS-CoV-2. Calu-3 cells were pre-treated with flagellin for 24 h before virus infection and then inoculated with SARS-CoV or SARS-CoV-2 at a low MOI and incubated for 48 h. Virus replication was determined by TCID50 titration of supernatants at indicated time points. As shown in Fig 3C, virus titers of SARS-CoV-2 and SARS-CoV were only marginally (twofold) increased at the earliest time points analyzed, and not affected at later time points in flagellin-treated versus control cells.

Overall, our data demonstrate that the mRNA expression of the virus-activating protease TMPRSS2 is up-regulated markedly by bacterial flagellin in Calu-3 human airway epithelial cells and to a

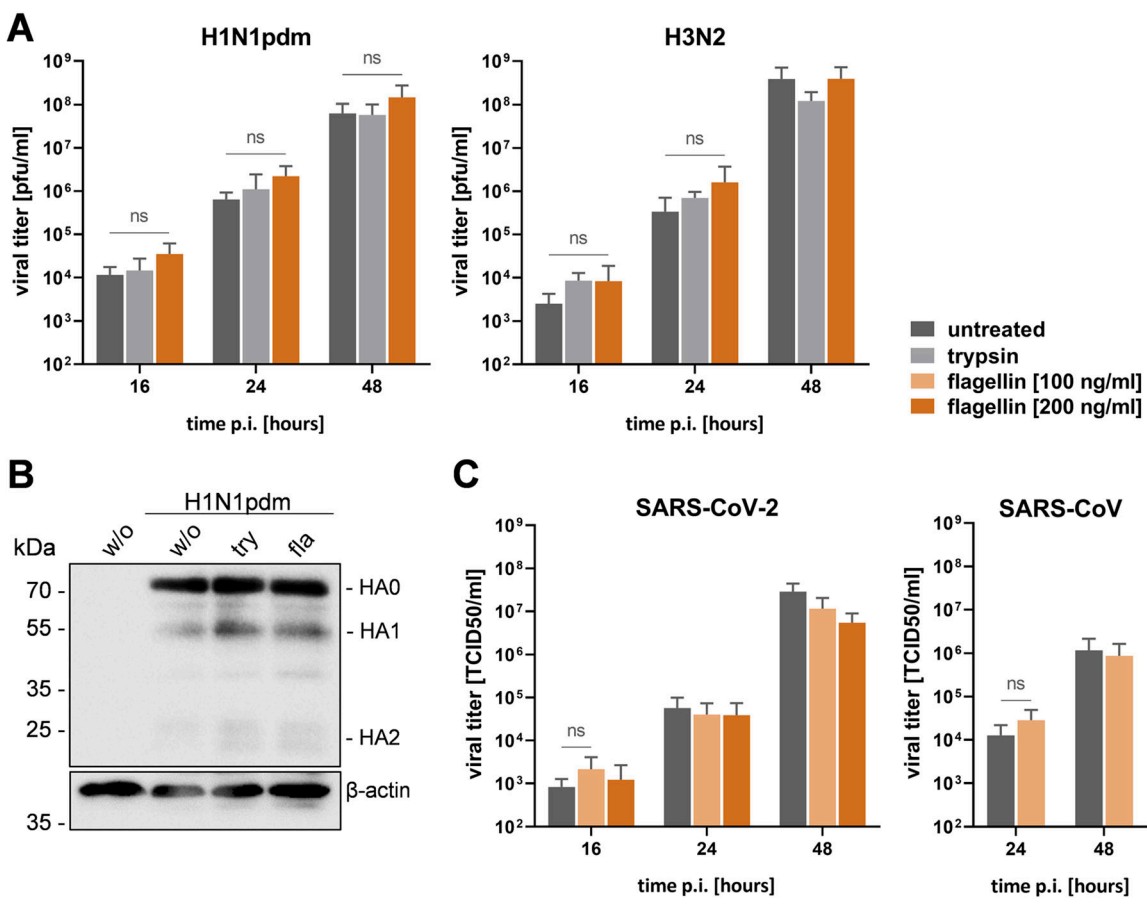

**Figure 3. Replication of IAV, SARS-CoV-2, and SARS-CoV in flagellin-treated airway cells.**
**(A)** Calu-3 cells were treated with flagellin (200 ng/ml) for 24 h, then infected with H1N1pdm or H3N2 IAV at a low MOI and incubated for 48 h in the absence or presence of trypsin. Virus titers in supernatants were determined by the plaque assay at the indicated time points. Data shown are mean values + SD of four independent experiments. **(B)** Calu-3 cells stimulated with flagellin (fla) for 24 h as described above were inoculated with H1N1pdm at an MOI of 0.001 and incubated for 48 h with or without exogenous trypsin (try). Cell lysates were subjected to SDS–PAGE and Western blot analysis using H1N1-specific antibodies. $\beta$-actin served as a loading control. **(C)** Calu-3 cells were incubated with flagellin for 24 h and then infected with SARS-CoV-2 or SARS-CoV at a low MOI. Virus titers were determined by TCID$_{50}$ titration at indicated time points. Data are mean values + SD (n = 3). Statistics: a $t$ test or Welch's $t$ test, ns $P > 0.05$ compared with the untreated infected control.

minor extent by LPS and Pam3Cys. The flagellin-induced increase in expression was specific for TMPRSS2 and not observed at this magnitude for other virus-activating proteases, namely, TMPRSS4, TMPRSS13, KLK5, matriptase, and furin. Virus titers of H1N1pdm and H3N2 IAV were increased up to 4.7-fold in flagellin-treated Calu-3 cells, whereas multicycle replication of SARS-CoV and SARS-CoV-2 was not affected. Thus, our data suggest that bacteria and in particular flagellated bacteria may support enhanced activation and replication of IAV in human airway cells via up-regulation of TMPRSS2 expression.

## Discussion

Co-infecting bacteria are a major cause of morbidity and mortality in influenza patients. The role of bacterial co-infection in the outcome of COVID-19 is less clear. The prevalence of co-infecting bacteria varied among COVID-19 patients in different studies ranging from 0.6% to 50% (Lai et al, 2020; Singh et al, 2021; Hedberg et al, 2022), but in general has been reported to be less common compared with influenza. Here, we demonstrate that bacteria, and particularly bacterial flagellin, increase TMPRSS2 expression in human airway cells, and thereby may enhance IAV activation and replication. Virus titers of H1N1pdm and H3N2 in Calu-3 cells were up to 3.4- and 4.7-fold increased, respectively, at different time points. Even though the difference in IAV replication in flagellin-treated and untreated cells was not striking, a threefold increase in virus load might significantly affect virus replication and spread in the respiratory tract. However, further investigation is needed to determine whether flagellated bacteria contribute to disease progression in co-infections of IAV and bacteria by up-regulating TMPRSS2 expression and consequently promoting HA cleavage. The most common co-infecting bacteria identified in influenza pneumonia, namely, *S. pneumoniae*, *S. aureus*, and *Haemophilus influenzae*, are non-flagellated, but also *L. pneumophila* or the flagellated bacterium *Pseudomonas aeruginosa* has been associated with co-infection in influenza patients and may even be

underestimated (Rizzo et al, 2010; Klein et al, 2016; Morris et al, 2017). A study performed during the 2009 H1N1 pandemic in Italy reported that 18% of patients hospitalized with influenza pneumonia showed co-infection with *L. pneumophila* (Rizzo et al, 2010). Chronic infection of *P. aeruginosa* in cystic fibrosis (CF) patients correlates with respiratory virus infections including IAV and influenza B viruses (Flight et al, 2014; Kiedrowski & Bomberger 2018). Importantly, we found that stimulation of Calu-3 cells with *S. pneumoniae*, LPS, or Pam3Cys also resulted in the significantly increased expression of TMPRSS2-mRNA. Thus, also non-flagellated bacteria may induce increased levels of TMPRSS2 in human airway cells and thereby promote enhanced virus activation. In contrast, IAV did not alter TMPRSS2-mRNA expression in Calu-3 cells (data not shown) and primary HBEC (Limburg et al, 2019).

Host cell proteases are critical determinants of viral tissue tropism and may contribute to virus pathogenicity. Interestingly, a study by Cheng et al demonstrated that single-nucleotide polymorphisms rs2070788 and rs383510 in the *TMPRSS2* gene predicted to induce higher levels of TMPRSS2 expression are associated with higher susceptibility to severe illness in patients with H1N1pdm and zoonotic H7N9 IAV infections (Cheng et al, 2015). Although experimental evidence for enhanced IAV activation and spread in the human airways because of *TMPRSS2* gene polymorphisms is missing, the study highlighted that elevated levels of TMPRSS2 might provide a predisposition for severe influenza outcomes. In COVID-19, the rs383510 polymorphism was associated with a two-fold increased infection risk for SARS-CoV-2, whereas a contribution to severe COVID-19 illness was discussed but remains to be demonstrated (Asselta et al, 2020; Schönfelder et al, 2021).

Numerous studies examined expression levels and tissue distribution of TMPRSS2 in comorbidities that have been associated as risk factors for severe COVID-19 outcomes. TMPRSS2 expression was found to be up-regulated in the lungs of smokers and COPD patients when compared to control tissues (Saheb Sharif-Askari et al, 2020; Fließer et al, 2021; Watson et al, 2021; Brake et al, 2022). Elevated expression levels of TMPRSS2 were also described in human pancreatic islets of obese donors compared with non-obese donors and in peripheral blood mononuclear cells (PBMC) of hypertension patients (Saheb Sharif-Askari et al, 2020; Taneera et al, 2020; Fließer et al, 2021). TMPRSS2 expression in the airways of hypertension patients or in obesity remains to be investigated. Patients with Down syndrome (also known as trisomy 21) are among the high-risk populations to develop severe COVID-19 (Emami et al, 2021). The *TMPRSS2* gene is located on chromosome 21 and was reported to show 60% higher levels of expression in Down syndrome and, thus, might contribute to the vulnerability of individuals with Down syndrome to severe COVID-19 disease (Toma & Dierssen, 2021). It remains to be demonstrated that increased TMPRSS2 expression supports enhanced SARS-CoV-2 activation and, as a consequence, a more severe COVID-19 outcome.

We found that replication of SARS-CoV-2 and SARS-CoV was not affected by flagellin treatment of Calu-3 cells. It remains to be further analyzed why multicycle replication of IAV but not CoV was increased upon flagellin stimulation in Calu-3 cells. Subcellular localization of virus activation by TMPRSS2 might contribute to the observed differences. Proteolytic activation of IAV HA and CoV S by TMPRSS2 differs in both compartmentalization and timing during

the viral life cycle. HA is cleaved intracellularly, most likely in the trans-Golgi network, during its transport to the plasma membrane (Böttcher-Friebertshäuser et al, 2010; Böttcher-Friebertshäuser, 2018). Priming of SARS-CoV-2 S is more complex compared with IAV HA and requires sequential cleavage by furin and TMPRSS2 at two distinct sites (Bestle et al, 2020; Hoffmann et al, 2020). Furin cleavage at the S1/S2 site occurs during egress of progeny virus and facilitates efficient binding to angiotensin-converting enzyme 2 (ACE2) that is used as a receptor upon entry into a new cell. Cleavage at the S2' site by TMPRSS2 immediately upstream of the fusion peptide finally primes S for membrane fusion and is believed to take place at the cell surface upon entry. ACE2 binding has been shown to be a prerequisite for exposure of the S2' site and subsequent cleavage by TMPRSS2 (Wrobel et al, 2020; Koch et al, 2021; Yu et al, 2022). Thus, enhanced CoV S cleavage by TMPRSS2 might require a simultaneous increase in ACE2 expression at the cell surface of the target cells or increased amounts of furin supporting elevated levels of S1/S2 site cleavage. We found that neither the mRNA expression of furin (Fig 1E) nor that of ACE2 (data not shown) was significantly altered by flagellin treatment of Calu-3 cells. In addition, the amount of glycoprotein present on the virion might provide a limiting factor for enhanced proteolytic activation. SARS-CoV-2 bears 20–40 S trimers at the surface, whereas IAV particles possess 340–400 HA trimers (Inglis et al, 1976; Harris et al, 2006; Turoňová et al, 2020). Thus, efficient activation of the 10-fold higher amount of HA might benefit from elevated TMPRSS2 levels, whereas protein basal levels of TMPRSS2 may already be sufficient to support proteolytic activation of the S incorporated in SARS-CoV-2 particles. Moreover, a flagellin-induced host response could suppress the replication of SARS-CoV and SARS-CoV-2 in Calu-3 cells. We observed that the titers of both viruses were even slightly reduced after 24 and 48 h p.i. in flagellin-treated cells compared with control cells.

Our data suggest that p38 MAPK/ERK- and NF-κB–dependent signaling pathways regulate the stimulation of TMPRSS2 expression upon flagellin stimulation. Moreover, recent studies lead us to assume that the transcription factor GATA2 is involved. The androgen-responsive expression of TMPRSS2 has been shown to be regulated by GATA2 in prostate cancer cells (Wang et al, 2007; Clinckemalie et al, 2013). GATA2 was found to be mediated by p38/ERK-dependent phosphorylation in acute myeloid leukemia cells and to be involved in the transcriptional regulation of several proinflammatory cytokines, including IL-1β (Katsumura et al, 2016). Recently, TMPRSS2 expression was shown to be stimulated by IL-1β and TNF-α via p38 MAPK/GATA2 signaling pathways in A549 cells and primary immortalized human alveolar epithelial cells (Cioccarelli et al, 2021). At present, it remains to be investigated whether flagellin-induced TMPRSS2 expression is also regulated by the transcription factor GATA2 together with NF-κB in Calu-3 cells.

The flagellin-induced increase in mRNA expression was specifically observed for TMPRSS2 and not at this magnitude for the other trypsin-like virus-activating proteases tested. Hence, the altered expression of TMPRSS2 upon flagellin stimulation may provide a hint for a physiological function of the protease, which is still unknown. TMPRSS2 is widely expressed in epithelial cells of the respiratory, gastrointestinal, and urogenital tract with high expression in the prostate and colon (Szabo & Bugge, 2008; Böttcher-

Friebertshäuser, 2018). TMPRSS2-deficient mice do not show an observable phenotype and develop normally (Kim et al, 2006). Knockout of TMPRSS2 activity is not compensated by the altered expression of closely related proteases in mice, suggesting functional redundancy. Alternatively, TMPRSS2 may exert a specialized function that is apparent only upon stress or disease. TMPRSS2 undergoes autocatalytic activation, indicating that it may initiate a proteolytic cascade (Afar et al, 2001). Our data suggest that TMPRSS2 might play a role in innate host response of airway cells to bacterial infections. We can only speculate that TMPRSS2 might be involved in zymogen activation of proteases such as matrix metalloproteinase-7 (MMP-7, matrilysin) or in processing of precursors of antimicrobial peptides (e.g., defensins). The hypothesis that TMPRSS2 may function in antimicrobial host response and inflammation in respiratory cells is strengthened by recent studies. The pro-inflammatory cytokines IL-1β and TNF-α were shown to induce TMPRSS2 expression in human lung A549 cells, which otherwise do not express significant levels of TMPRSS2. In contrast, IL-6 did not stimulate TMPRSS2 expression in A549 cells (Cioccarelli et al, 2021). In accordance with this, we did not observe an increase in TMPRSS2-mRNA expression upon IL-6 treatment of Calu-3 cells (data not shown). Treatment of primary basal nasal cells with the antimicrobial and anti-inflammatory drug azithromycin significantly down-regulated the expression of TMPRSS2 and the related virus-activating protease TMPRSS11D/HAT (Renteria et al, 2020). Interestingly, Iwata-Yoshikawa et al found that TMPRSS2-knockout mice showed weakened inflammatory cytokine responses in the lungs upon intranasal stimulation with the dsRNA mimetic poly(I:C), indicating that TMPRSS2 may be involved in the expression or release of different cytokines (Iwata-Yoshikawa et al, 2019). In agreement with this, a recent study showed that TMPRSS2 impacts cytokine expression in murine dendritic cells (Gunne et al, 2023).

During preparation of this article, Ruffin et al published a study that demonstrated that *P. aeruginosa* increases the expression of TMPRSS2 in primary airway epithelial cells of cystic fibrosis (CF) patients (Ruffin et al, 2021). In agreement with our data, they showed that flagellin is the dominant structural component of *P. aeruginosa* that stimulates this effect and that flagellin increases TMPRSS2 expression in Calu-3 cells deficient for the ion channel transmembrane conductance regulator (CFTR) in a TLR5- and p38 MAPK-dependent manner. CFTR-deficient Calu-3 cells were more susceptible to SARS-CoV-2 infection compared with wild-type cells independent of flagellin stimulation. However, in agreement with our results, flagellin treatment did not affect SARS-CoV-2 virus titers in the supernatants. Taken together, the data by us and Ruffin and co-workers demonstrate that flagellated bacteria can stimulate the increased expression of TMPRSS2 in human airway cells.

TMPRSS2 has been established as a major activating protease of IAV and various CoV and may be critical for proteolytic activation of other respiratory viruses as well (Böttcher-Friebertshäuser, 2018; Bestle et al, 2020, 2021). During the COVID-19 pandemic, a number of protease inhibitor candidates were shown to provide potent antiviral activity against SARS-CoV-2 in human airway cells and in animal models in vivo and hence are worth further drug development (Bestle et al, 2020; Mahoney et al, 2021; Shapira et al, 2022). Moreover, the broad-range serine protease inhibitor aprotinin has been demonstrated to be effective against COVID-19 in a phase III

clinical study using an aerosol application without causing side effects (Redondo-Calvo et al, 2022).

Overall, our data suggest that TMPRSS2 expression in airway cells is elevated upon bacterial infection. Whether this contributes to virus activation and disease outcome in co-infections of IAV and bacteria remains open. In addition, our data indicate that TMPRSS2 has a physiological function in antimicrobial host response of airway epithelial cells, and particularly in host response against bacteria.

# Materials and Methods

## Cells

The human airway epithelial cell line Calu-3 (HTB55; ATCC) was propagated in DMEM–Ham's F-12 (1:1) (Gibco) supplemented with 10% FCS, glutamine, and antibiotics. Every 3–4 d, the culture medium was renewed. Cell growth and incubations were performed at 37°C and 5% $CO_2$. Madin–Darby canine kidney II (MDCK(II)) cells were maintained in DMEM (Gibco) supplemented with 10% FCS, antibiotics, and glutamine. Primary HBEC obtained from Lonza were cultivated in 24-well plates coated with 0.05 mg collagen from calfskin (Sigma-Aldrich) in the ready-to-use airway epithelial cell growth medium (AEGM; PromoCell) supplemented with antibiotics.

## Viruses

IAV used in this study were A/Hamburg/5/09 (H1N1pdm) and recombinant A/Hong Kong/1/68 (H3N2) virus (kindly provided by Mikhail Matrosovich, Institute of Virology, Marburg, Germany). IAV were propagated on MDCK(II) cells in DMEM supplemented with 0.1% BSA, glutamine, and antibiotics (infection medium) supplemented with 1 μg/ml TPCK-treated trypsin (Sigma-Aldrich). To harvest virus stock from infected cells, the supernatant was collected at 72 h p.i., cleared by low-speed centrifugation, and stored at −80°C.

SARS-CoV-2 (isolate Germany/Wetzlar/2020, wild type) and SARS-CoV (isolate FFM-1, GenBank accession number AY310120) were used. All experiments with SARS-CoV-2 were conducted under BSL-3 conditions, and experiments with SARS-CoV were conducted under BSL-4 conditions. SARS-CoV-2 and SARS-CoV were propagated on Calu-3 cells in DMEM supplemented with 3% FCS, glutamine, and antibiotics. To harvest virus stock, the supernatant was collected from infected cells at 72–96 h p.i. and stored at −70°C (SARS-CoV) or −80°C (SARS-CoV-2) after low-speed centrifugation.

## Bacteria

*L. pneumophila* strains Corby wild type (WT) and flagellin-deficient mutant (Δfla) have been described previously (Schmeck et al, 2008). *L. pneumophila* were routinely cultured on buffered charcoal–yeast extract (BCYE) Agar plates at 37°C. *S. pneumoniae* strain D39 (Schmeck et al, 2004) was grown on BD Columbia Agar plates with 5% sheep blood (Becton Dickinson) overnight and then further

propagated in Todd–Hewitt Broth (THY; Carl Roth) supplemented with 0.5% yeast extract (Carl Roth).

### RNA isolation and RT–qPCR

Total RNA isolation was performed using Monarch Total RNA Miniprep Kit (NEB) according to the manufacturer's protocol. In order to analyze mRNA expression levels of different proteases, RT–qPCR was performed using Luna Universal One-Step RT–qPCR Kit (NEB) according to the supplier's protocol. Primer sequences are available upon request. The data were quantified according to the $2^{-\Delta\Delta Ct}$ method with transcripts of the housekeeping gene $\alpha$-tubulin used for normalization.

### Bacterial infection of Calu-3 cells

Calu-3 cells cultivated in 24-well plates until confluency were inoculated with *L. pneumophila* or *S. pneumoniae* in the antibiotics-free infection medium at an MOI of 100 or 0.5, respectively. To stimulate the cells with heat-inactivated *L. pneumophila*, the bacteria were incubated at 95°C for 10 min before inoculation. After 24-h incubation at 37°C, the cells were washed with PBS before the RNA was isolated and subjected to RT–qPCR.

### Stimulation of Calu-3 cells and HBEC

Confluent Calu-3 cells grown in 24-well plates were washed once and then stimulated with 100–200 ng/ml standard flagellin isolated from *Salmonella typhimurium* (InvivoGen), 5 µg/ml Pam3Cys (InvivoGen), or 1 µg/ml LPS (Sigma-Aldrich) diluted in the infection medium for 24 h at 37°C. Subsequently, the cells were washed with PBS. Total RNA was harvested as described above and subjected to RT–qPCR. Treatment with 10 µM ERK inhibitor U0126 monoethanolate (U120; Merck), p38 MAPK inhibitor SB 202190 (S7067; Merck), or NF-κB inhibitor BAY 11-7821 (Merck Millipore) was performed for 1 h before flagellin stimulation. To analyze the expression of the TMPRSS2 protein, cells were treated with flagellin (200 ng/µl) for 24 h or remained untreated. Subsequently, cells were subjected to SDS–PAGE and Western blot analysis as described below.

Primary HBEC cultivated in 24-well plates were treated with flagellin diluted in the ALI-Mix medium (1:1 mixture of DMEM (Sigma-Aldrich) and AEGM, supplemented with 60 ng/ml retinoic acid) at the indicated concentrations for 4 or 24 h.

### Virus infection of cells and multicycle viral replication in the presence of flagellin

For analysis of IAV replication kinetics, Calu-3 cells were grown to near confluency in 24-well plates. Cells were treated with 200 ng/ml flagellin for 24 h or remained untreated. After removal of flagellin, cells were washed with PBS and inoculated with IAV in the infection medium at an MOI of 0.001 for 1 h, washed with PBS, and further incubated in the fresh infection medium in the absence or presence of 0.5 µg/ml trypsin. Supernatants were collected at 16, 24, and 48 h p.i. and used to determine virus titers by the plaque assay in MDCK(II) cells with the Avicel overlay as described previously

(Limburg et al, 2019). To analyze HA cleavage, Calu-3 cells treated with flagellin and infected in the absence or presence of trypsin as described above were harvested 48 h p.i. Cells were then subjected to SDS–PAGE and Western blot analysis.

For infection with SARS-CoV-2 or SARS-CoV, Calu-3 cells were cultivated in 12-well plates. At 90–100% confluency, cells were treated with flagellin (100–200 ng/ml) for 24 h or remained untreated. The cells were then washed with PBS and inoculated under serum-free conditions with SARS-CoV-2 or SARS-CoV at an MOI of 0.00001 or 0.001, respectively, for 1 h at 37°C and 5% $CO_2$. Subsequently, the cells were washed again with PBS and were further incubated in DMEM supplemented with 3% FCS, glutamine, and antibiotics. The supernatant was collected at several time points p.i., and virus titers were determined by 50% tissue culture infectious dose ($TCID_{50}$) titration. $TCID_{50}$ titration of SARS-CoV-2 supernatants was performed by serially diluting the supernatants from $5^{-1}$ to $5^{-11}$ in serum-free DMEM supplemented with glutamine and antibiotics in technical quadruplicates as described previously (Bestle et al, 2020). Virus dilutions were transferred to Calu-3 cells grown in 96-well plates and incubated for 72 h. Virus titers were determined using the Spearman and Kärber algorithm.

### SDS–PAGE and Western blot analysis

Cells were washed in PBS, lysed in reducing SDS–PAGE sample buffer, and heated at 95°C for 15 min. Samples were subjected to SDS–PAGE (12% gel) and transferred to a polyvinylidene difluoride (PVDF) membrane (GE Healthcare). Subsequently, proteins were detected by incubation with primary antibodies and species-specific peroxidase-conjugated secondary antibodies (DAKO). Proteins were visualized using the ChemiDoc XRS+ system with Image Lab software (Bio-Rad). A polyclonal rat serum against TMPRSS2 was generated by DNA vaccination (Genovac/Aldevron). A polyclonal rabbit serum against HA of H1N1 was obtained from Sino Biological (11085-T62). A monoclonal mouse anti-$\beta$-actin antibody was purchased from Abcam (ab6276).

### Data analysis

Data are shown as mean values + SD for at least three biologically independent experiments. All graphs were done using GraphPad Prism (version 9), and statistical analysis was performed using R (version 3.6.1). Levene's test was performed to assess the homogeneity of variance. A *t* test or Welch's *t* test for unpaired samples was performed to analyze data sets with or without the homogeneity of variance, respectively. Differences were considered significant at $P < 0.05$.

## Data Availability

Data supporting the findings of this study are available within the article or if stated otherwise available from the corresponding author (Eva Böttcher-Friebertshäuser, friebertshaeuser@staff.uni-marburg.de) upon reasonable request.

**Life Science Alliance**

# Supplementary Information

# Acknowledgements

This research was in part funded by the German Research Foundation (DFG), KFO309 Project P6, 284237345 to E Böttcher-Friebertshäuser and (DFG, SFB-TR84, C1) to B Schmeck; Universities of Giessen and Marburg (UKGM) Research Funding to E Böttcher-Friebertshäuser; Bundesministerium für Bildung und Forschung (e:Med CAPSYS-FKZ 01ZX1604E, ERACoSysMed2-SysMed-COPD-FKZ 031L0140) to B Schmeck; and Hessisches Ministerium für Wissenschaft und Kunst (LOEWE Diffusible Signals) to AL Jung and B Schmeck.

## Author Contributions

M Schwerdtner: data curation, formal analysis, validation, investigation, visualization, methodology, and writing—review and editing.
A Skalik: data curation, formal analysis, validation, investigation, visualization, methodology, and writing—review and editing.
H Limburg: validation, investigation, methodology, and writing—review and editing.
J Bierwagen: validation, investigation, and methodology.
AL Jung: conceptualization, resources, formal analysis, funding acquisition, validation, methodology, and writing—review and editing.
J Dorna: resources, validation, investigation, and methodology.
A Kaufmann: conceptualization, resources, validation, methodology, and writing—review and editing.
S Bauer: conceptualization, resources, and writing—review and editing.
B Schmeck: conceptualization, resources, funding acquisition, validation, methodology, and writing—review and editing.
E Böttcher-Friebertshäuser: conceptualization, resources, data curation, formal analysis, supervision, funding acquisition, validation, investigation, methodology, project administration, and writing—original draft, review, and editing.

## Conflict of Interest Statement

The authors declare that they have no conflict of interest.

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
