## [Reviewer comments · Life Science Alliance]

Life Science Alliance

Expression of TMPRSS2 is upregulated by bacterial flagellin, LPS and Pam3Cys in human airway cells

Marie Schwerdtner, Annika Skalik, Hannah Limburg, Jeff Bierwagen, Anna Lena Jung, Jens Dorna, Andreas Kaufmann, Stefan Bauer, Bernd Schmeck, and Eva Böttcher-Friebertshäuser

DOI: <https://doi.org/10.26508/lsa.202201813>

Corresponding author(s): *Eva Böttcher-Friebertshäuser, Philipps-Universität Marburg*

Review Timeline:

Submission Date:	2022-11-09
Editorial Decision:	2022-12-16
Revision Received:	2023-03-15
Editorial Decision:	2023-04-12
Revision Received:	2023-05-09
Editorial Decision:	2023-05-09
Revision Received:	2023-05-09
Accepted:	2023-05-10

Scientific Editor: Novella Guidi

Transaction Report:

December 16, 2022

Re: Life Science Alliance manuscript #LSA-2022-01813

Prof. Eva Böttcher-Friebertshäuser
Philipps-Universität Marburg
Institute of Virology
Hans-Meerwein-Str. 2
Marburg 35043
Germany

Dear Dr. Böttcher-Friebertshäuser,

Thank you for submitting your manuscript entitled "Expression of TMPRSS2 is upregulated by bacterial flagellin, LPS and Pam3Cys in human airway cells" to Life Science Alliance. The manuscript was assessed by expert reviewers, whose comments are appended to this letter. We invite you to submit a revised manuscript addressing the Reviewer comments.

Thank you for this interesting contribution to Life Science Alliance. We are looking forward to receiving your revised manuscript.

Sincerely,

B. MANUSCRIPT ORGANIZATION AND FORMATTING:

Reviewer #1 (Comments to the Authors (Required)):

It is well known that bacterial co-infection can cause severe influenza. It has also been shown that the host protease TMPRSS2 is important for the activation of influenza virus HA proteins. Therefore, Schwerdtner et al. used a cultured cell model (Calu3 cells) to analyze whether bacterial infection affects TMPRSS2 expression. The results showed that *Legionella pneumophila* flagellin (and, to a lesser extent, other bacterial-derived components) increased TMPRSS2 mRNA levels via the p38 MAPK signaling pathway and promoted influenza virus and coronavirus multiplication.

The topic covered is very interesting and important. The conclusions the authors argue are also very fascinating. However, the following points need to be considered

One of the most serious concerns in this study is whether the small differences in changes in viral proliferation are statistically significant. The level of change is so small that statistical significance must be demonstrated from multiple angles. In addition, additional virological analyses would be needed to support the conclusions. For example, the following points need to be considered

Does the increased ability of influenza viruses to multiply with increased expression of TMPRSS2 imply that the activation level of influenza viruses grown in untreated Calu-3 is incomplete? If so, why is its activation not improved by trypsin? Also, at least the level of HA cleavage should be evaluated at the protein level and data should be presented, but no such data has been presented.

Reviewer #2 (Comments to the Authors (Required)):

This manuscript addresses the interesting and important question of how co-infecting bacteria may affect influenza infection through the action of HA-activating proteases. The authors follow up an observation from the 2009 pandemic where by *Legionella* co-infections were noted. The main focus of the paper is to provide evidence regarding whether both *Legionella*, or its flagellin protein, upregulates TMPRSS2 activity (as compared to TMPRSS4 and IL-6), and so upregulates viral infection of Calu-3 cells

The data are interesting, but still somewhat preliminary

Figure 1 shows increased levels of mRNA for the proteins in question, but no expression or activity of the proteases/proteins themselves. This is a notable limitation of the study.

In Figure 1 it is hard to determine the true biological effect. While statistical significance is shown in many cases, it is hard (as currently presented) to determine what is being compared to what with regard to the p-values noted with asterisks. It may be better to re-plot to group the proteases/proteins together, rather than to group the infectious agents added together, and to make use of brackets to describe the where the asterisks are being used.

In Figure 2, the authors address the effect of adding flagellin to infections mediated by H1N1 H3N3 SARS-CoV-2, and SARS-CoV-1 (in comparison to adding trypsin). While there appears to be some effect of flagellin, especially at early time point, no statistical analysis is provided, so it remains unclear on any possible biological relevance.

Reviewer #3 (Comments to the Authors (Required)):

This study described impact of bacteria infection on the expression of TMPRSS2, a host factor required by many virus infection. Specifically, they showed that pneumophila flagellin, LPS and Pam3Cys can stimulate the expression of TMPRSS2, which can be reduced by inhibiting MAPK/ERK and NF- κ B. Relation between virus and bacteria infections is of great importance for

developing treatment to co-infection. Here are questions for author:

1. The author utilised the mutant delta-flagellin to identify flagellin as the dominant component enhancing Tmprss2 expression. However this mutant seems not as infectious as the wild type bacteria, which only stimulated weaker immune response (fig 1A). Is that possible the lower Tmprss2 expression (fig 1A, delta-fla) was caused by the impaired infection or weaker immune response? Can the author provide more explanation or evidence to show that flagellin is the major factor leading to Tmprss2 up-regulation?
2. Will the bacteria induced Tmprss2 up-regulation also occur in other human lung cells, for example, the alveolar epithelial cells? If the author observe the same phenomena in different cell line, the study will be more complete and significant.
3. Tmprss2 is critical for virus membrane fusion and viral entry. Elevated expression of it may enhance cell-cell fusion or cell-cell transmission. In the virus infection assay (fig 2), is there any difference in syncytial formation between the groups with and without flagellin treatment?
4. In the virus infection assay, will virus replication be different if the cells are pre-infected with bacteria? Fig 2 showed no difference of viral titer between groups with and without flagellin. Is that because flagellin only is not sufficient? Can the author add another group with bacteria infection prior to virus infection as a positive control? For example, the cells can be exposed to bacteria for 1 day, followed with removing the medium, washing and adding virus.

Reviewer #1

The topic covered is very interesting and important. The conclusions the authors argue are also very fascinating. However, the following points need to be considered

- One of the most serious concerns in this study is whether the small differences in changes in viral proliferation are statistically significant. The level of change is so small that statistical significance must be demonstrated from multiple angles.

Statistical analysis of viral titers was performed using Student's t-test or Welch's t-test. Differences were considered significant at $P < 0.05$. This is noted in the figure legend of Fig. 3 and in the material and methods section (lines 536-539). In all our experiments, an increase in viral titer was detected in flagellin- and partially trypsin-treated cells, but the differences were not significant. We have additionally marked this in the graphs (Fig. 3) in the revised manuscript. We also address this point in the discussion section (lines 301-304).

We also observed an increase in TMPRSS2-mRNA expression and a 3-fold increase in H1N1pdm viral titer in primary human bronchial cells. The new data are shown in the revised manuscript (Fig. 3C, text lines 206-216).

Although we cannot show significant differences, we believe that a 3 to 4-fold increase in virus titer in patients may have an impact on the course of the disease and might contribute to the severity of co-infections.

- Does the increased ability of influenza viruses to multiply with increased expression of TMPRSS2 imply that the activation level of influenza viruses grown in untreated Calu-3 is incomplete? If so, why is its activation not improved by trypsin?

We and other groups actually observe that the HA0 of human influenza viruses is never completely cleaved in human cells (Calu-3, Caco-2 as well as primary human bronchial or alveolar epithelial cells). Often, approximately 50% cleaved and 50% un-cleaved HA is detected. HA cleavage can be increased in percentage by trypsin, but this does not usually cause a very large increase in virus titer. This indicates that by far not all HA of a virus particle has to be cleaved and that cleavage by endogenous proteases is sufficient for virus infectivity. We have addressed this aspect in the revised manuscript (lines 261-265) and examined HA cleavage for H1N1 in Fig. 3B.

- Also, at least the level of HA cleavage should be evaluated at the protein level and data should be presented, but no such data has been presented.

We have added data on HA cleavage for H1N1pdm in Calu-3 cells as Fig. 3B.

Reviewer #2

The data are interesting, but still somewhat preliminary.

- Figure 1 shows increased levels of mRNA for the proteins in question, but no expression or activity of the proteases/proteins themselves. This is a notable limitation of the study.

Thank you for the critical comment - we agree with that. It is not easy to detect endogenous TMPRSS2 at the protein level, because TMPRSS2 is usually very weakly expressed in airway cells. After many attempts, we succeeded. Figure 1B shows that TMPRSS2 expression is also increased at the protein level when Calu-3 cells are stimulated with flagellin.

- In Figure 1 it is hard to determine the true biological effect. While statistical significance is shown in many cases, it is hard (as currently presented) to determine what is being compared to what with regard to the p-values noted with asterisks. It may be better to re-plot to group the proteases/proteins together, rather than to group the infectious agents added together, and to make use of brackets to describe the where the asterisks are being used.

We have redrawn the diagram accordingly. We also rearranged the diagrams in Figs. 1C and 1D in the same way.

- In Figure 2, the authors address the effect of adding flagellin to infections mediated by H1N1 H3N3 SARS-CoV-2, and SARS-CoV-1 (in comparison to adding trypsin). While there appears to be some effect of flagellin, especially at early time point, no statistical analysis is provided, so it remains unclear on any possible biological relevance.

Statistical analysis of viral titers was performed using Student's t-test or Welch's t-test. Differences were considered significant at $P < 0.05$. This is noted in the figure legend of Fig. 3 and in the material and methods section (lines 536-539). In all our experiments, an increase in viral titer was detected in flagellin- and partially trypsin-treated cells, but the differences were not significant. We have additionally marked this in the graphs (Fig. 3) in the revised manuscript. We also address this point in the discussion section (lines 301-304).

We also observed an increase in TMPRSS2-mRNA expression and a 3-fold increase in H1N1pdm viral titer in primary human bronchial cells. The new data are shown in the revised manuscript (Fig. 3C, text lines 206-216).

Although we cannot show significant differences, we believe that a 3 to 4-fold increase in virus titer in patients may have an impact on the course of the disease and might contribute to the severity of co-infections.

Reviewer #3

1.The author utilised the mutant delta-flagellin to identify flagellin as the dominant component enhancing TMPRSS2 expression. However, this mutant seems not as infectious as the wild type bacteria, which only stimulated weaker immune response (fig 1A).

- Is that possible the lower TMPRSS2 expression (fig 1A, delta-fla) was caused by the impaired infection or weaker immune response? Can the author provide more explanation or evidence to show that flagellin is the major factor leading to TMPRSS2 up-regulation?

We cannot completely exclude the possibility that the lower stimulation of the immune response by the delta-fla mutant also had some effect on TMPRSS2 expression. However, since we can induce a strong increase in TMPRSS2 expression using recombinant flagellin, whereas LPS stimulates TMPRSS2 expression less, we think our data show that flagellin is the dominant component. The slight but significant increase in TMPRSS2-expression observed in delta-fla stimulated Calu-3 cells is most likely due to LPS.

2. Will the bacteria induced TMPRSS2 up-regulation also occur in other human lung cells, for example, the alveolar epithelial cells? If the author observe the same phenomena in different cell line, the study will be more complete and significant.

Thank you for the comment. Calu-3 cells are derived from lung tissue. To examine whether TMPRSS2 expression is stimulated by flagellin also in other respiratory cells we determined expression of TMPRSS2-mRNA in primary human nasal and bronchial epithelial cells (HNEC, HBEC) upon flagellin stimulation. Flagellin treatment resulted in increased TMPRSS2-mRNA expression in HBEC and 3-fold increased H1N1pdm titers at early time points. The data are shown in new Fig. 1G and Fig. 3C of the revised manuscript.

Primary nasal cells (HNEC) from two different donors responded differently to treatment with flagellin, and in some cultures we did not observe an increase in IL-6 mRNA after stimulation with flagellin. Other HNEC cultures showed a ca. 10-fold increase in IL-6 mRNA and a 2-fold increase in TMPRSS2-mRNA expression at 24 h post flagellin stimulation. Due to these differences, we have not included the data on nasal cells in the manuscript.

3. TMPRSS2 is critical for virus membrane fusion and viral entry. Elevated expression of it may enhance cell-cell fusion or cell-cell transmission. In the virus infection assay (fig 2), is there any difference in syncytial formation between the groups with and without flagellin treatment?

We have not observed any differences in the formation of syncytia. For influenza A viruses we did not expect any syncytia. Also for SARS-CoV and SARS-CoV-2 we could not see clear syncytia in Calu-3 cells. 24-48 h after infection, the cells show a clear CPE in the form of rounded cells. We did not observe a difference in the CPE in flagellin-treated versus untreated cells for SARS-CoV and SARS-CoV-2.

4. In the virus infection assay, will virus replication be different if the cells are pre-infected with bacteria? Fig 2 showed no difference of viral titer between groups with and without flagellin. Is that because flagellin only is not sufficient? Can the author add another group

with bacteria infection prior to virus infection as a positive control? For example, the cells can be exposed to bacteria for 1 day, followed with removing the medium, washing and adding virus.

Thank you for the comment. At the beginning of the project, we conducted co-infections of influenza viruses with legionella. However, pre-inoculation of the cells with legionella led to a worse initial infection with influenza viruses, so that we switched to recombinant flagellin.

Legionella may also be a difficult model in this case, as it primarily replicates in alveolar macrophages and not in epithelial cells. Presumably, co-culture models of airway cells and macrophages would be a suitable model for further investigation. This is beyond the scope of the current manuscript, but we may consider it for future studies and try to establish such models.

April 12, 2023

Re: Life Science Alliance manuscript #LSA-2022-01813R

Prof. Eva Böttcher-Friebertshäuser
Philipps-Universität Marburg
Institute of Virology
Hans-Meerwein-Str. 2
Marburg 35043
Germany

Dear Dr. Böttcher-Friebertshäuser,

Thank you for submitting your revised manuscript entitled "Expression of TMPRSS2 is upregulated by bacterial flagellin, LPS and Pam3Cys in human airway cells" to Life Science Alliance. The manuscript has been seen by the original reviewers whose comments are appended below. While one reviewer continues to be overall positive about the work in terms of its suitability for Life Science Alliance, Reviewer 1 still claims that important issues remain, that were previously shared by Reviewer 2 as well.

Please address Reviewer 1's comments carefully paying close attention to results that do not add to statistical significance. If a result is not significantly different then there is no biological relevance, so please tone down your conclusions accordingly. Also, we encourage you to add biological replicates to the experiments that do not have at least 3 replicates (e.g. Fig 3C which has only 2).

Our general policy is that papers are considered through only one revision cycle; however, we are open to one additional short round of revision. Please note that I will expect to make a final decision without additional reviewer input upon resubmission.

Please submit the final revision within one month, along with a letter that includes a point by point response to the remaining reviewer comments.

To upload the revised version of your manuscript, please log in to your account: <https://lsa.msubmit.net/cgi-bin/main.plex>
You will be guided to complete the submission of your revised manuscript and to fill in all necessary information.

- A letter addressing the reviewers' comments point by point.
- An editable version of the final text (.DOC or .DOCX) is needed for copyediting (no PDFs).
- High-resolution figure, supplementary figure and video files uploaded as individual files: See our detailed guidelines for preparing your production-ready images, <https://www.life-science-alliance.org/authors>
- Summary blurb (enter in submission system): A short text summarizing in a single sentence the study (max. 200 characters including spaces). This text is used in conjunction with the titles of papers, hence should be informative and complementary to the title and running title. It should describe the context and significance of the findings for a general readership; it should be written in the present tense and refer to the work in the third person. Author names should not be mentioned.

B. MANUSCRIPT ORGANIZATION AND FORMATTING:

Sincerely,

Reviewer #1 (Comments to the Authors (Required)):

Unfortunately, this is not a sufficient response to the primary comments by this reviewer and only partially addresses this reviewer's concerns.

As mentioned in the previous review comments, the most disturbing aspect of the content of this paper is that it is not clear whether the small fluctuations in values are within experimental error or are significant differences; there are additional data that reinforce that the increase in TMPRSS2 mRNA is a significant change. On the other hand, there is still insufficient scientific data to draw conclusions about viral replication.

Even though the authors state in the text that viral replication has increased several fold, they likewise state in the text that the increase is not statistically significant. If a statistically significant increase cannot be confirmed, then scientifically no increase should be confirmed. In fact, in both Figures 3A and 3C, there is little statistically significant difference in the amount of viral increase. Despite the reviewers' comments regarding concerns about statistical significance, the revised manuscript does not provide sufficient additional details about what statistical treatment was used in the repeated experiments. As a result, as far as this reviewer can judge from the data presented here, this reviewer has to raise concerns about the conclusions that can be drawn from the data in the manuscript.

Furthermore, we have not received a response to the second comment requested by this reviewer. The authors state that in Calu3 cells, flagellin treatment promotes TMPRSS2 expression, resulting in increased HA cleavage and thus increased viral replication. If so, one would expect trypsin treatment to also increase viral replication, but trypsin treatment does not lead to increased viral replication, even though it promotes HA cleavage as well. The increase in viral replication by flagellin treatment was very small (hardly significant), and the possibility that it was all within the experimental margin of error cannot be ruled out, which does not at all resolve this reviewer's concerns.

Reviewer #2

most of my former concerns have been addressed, the upregulation TMPRSS2 is now clear, as well as the effects on influenza infection - however the lack of effect on SARS-CoV-1 and SARS-CoV-2 is puzzling as these viruses should also be influenced by TMPRSS2 levels - the authors do attempt to explain this difference, but perhaps the current paper is more suited to simply reporting the influenza data until the coronavirus results can be better explained

Reviewer #3 (Comments to the Authors (Required)):

The resubmitted manuscript has been largely improved regarding the completeness. The revision has addressed my questions adequately. No further questions are to the authors.

Editor's advice:

Please address Reviewer 1's comments carefully paying close attention to results that do not add to statistical significance. If a result is not significantly different then there is no biological relevance, so please tone down your conclusions accordingly. Also, we encourage you to add biological replicates to the experiments that do not have at least 3 replicates (e.g. Fig 3C which has only 2).

We repeated all experiments in primary HBEC cells. In the revised manuscript we can demonstrate that TMPRSS2-mRNA expression is significantly increased by flagellin treatment in primary HBEC (Fig. 1G, n=3 independent experiments).

We also repeated the growth curves of H1N1pdm virus in primary HBEC (see Fig. 3C). Unfortunately, the primary cells (which were from a different donor) were less susceptible to infection this time, and virus titers were generally low in both flagellin-treated and untreated cells. We therefore decided not to include the data on H1N1pdm replication in HBEC in the final manuscript due to the low number of biological replicates and some variation in viral titers.

Reviewer 1:

As mentioned in the previous review comments, the most disturbing aspect of the content of this paper is that it is not clear whether the small fluctuations in values are within experimental error or are significant differences. In fact, in both Figures 3A and 3C, there is little statistically significant difference in the amount of viral increase. ... This reviewer has to raise concerns about the conclusions that can be drawn from the data in the manuscript.

We understand the reviewer's criticism and have toned down our conclusions accordingly in the revised manuscript (e.g. Abstract, Summary blurb, Discussion: lines 295-298, 417-418). We still conclude that flagellin/bacteria-induced increase in TMPRSS2 activity may play a role in virus activation and replication in influenza virus coinfections, but clarify that our results do not allow us to draw conclusions on whether this contributes to disease progression in co-infections.

We think that the manuscript in this revised version now also meets the criticisms of reviewer 1, who had initially also been very positive about the manuscript and the conclusions. ("The topic covered is very interesting and important. The conclusions the authors argue are also very fascinating.")

Furthermore, we have not received a response to the second comment requested by this reviewer. The authors state that in Calu3 cells, flagellin treatment promotes TMPRSS2 expression, resulting in increased HA cleavage and thus increased viral replication. If so, one would expect trypsin treatment to also increase viral replication, but trypsin treatment does not lead to increased viral replication, even though it promotes HA cleavage as well.

- We observe a slight increase in viral titers at early time points by trypsin treatment. We have described this more clearly in the results section of the revised manuscript (lines 257-259).

- It is important to note that treatment with trypsin does not necessarily lead to a significant increase in viral titer. In cells lacking appropriate endogenous virus-activating proteases, trypsin is essential for virus replication. In contrast, several studies show that the addition of trypsin to cells expressing endogenous trypsin-like proteases has a smaller effect on viral titer. This is presumably because activation by endogenous proteases is already sufficient. We had added this point and some references in the first revision of the manuscript (Results: lines 260-265).

Reviewer 2:

Most of my former concerns have been addressed, the upregulation TMPRSS2 is now clear, as well as the effects on influenza infection - however the lack of effect on SARS-CoV-1 and SARS-CoV-2 is puzzling as these viruses should also be influenced by TMPRSS2 levels ...

Activation of the CoV spike protein is more complex than activation of the influenza virus HA (two cleavage sites versus one cleavage site) and different proteases can be involved. It is not yet known whether more TMPRSS2 expression actually means increased virus activation. This has been proposed by a number of studies but evidence is still missing.

... the authors do attempt to explain this difference, but perhaps the current paper is more suited to simply reporting the influenza data until the coronavirus results can be better explained.

We would like to leave the data on SARS-CoV and SARS-CoV-2 in the manuscript, even if we cannot conclusively explain why the titers of both CoV are not increased despite increased TMPRSS2 expression. In our view, this is an important and interesting observation, which also relates to work by Ruffin et al. 2021: Ruffin and colleagues observed in cells from patients with cystic fibrosis that the proliferation and release of SARS-CoV-2 was not increased despite increased TMPRSS2 expression. In their study, the authors discuss an immune response as a possible cause. We discuss the data by Ruffin et al. in our manuscript (Discussion: lines 398-408) and also included the possibility of an flagellin-induced immune response-mediated reduction in CoV titers (Discussion: lines 357-359). We observe that the CoV titers in flagellin-treated cells are reduced after 24 and 48 h compared to control cells, which may well be the result of a flagellin-induced immune response of the cells.

May 9, 2023

RE: Life Science Alliance Manuscript #LSA-2022-01813RR

Prof. Eva Böttcher-Friebertshäuser
Philipps-Universität Marburg
Institute of Virology
Hans-Meerwein-Str. 2
Marburg 35043
Germany

Dear Dr. Böttcher-Friebertshäuser,

Thank you for submitting your revised manuscript entitled "Expression of TMPRSS2 is upregulated by bacterial flagellin, LPS and Pam3Cys in human airway cells". We would be happy to publish your paper in Life Science Alliance pending final revisions necessary to meet our formatting guidelines.

- please add the author contributions to the main manuscript text
- please use the [10 author names, et al.] format in your references (i.e. limit the author names to the first 10)

A. FINAL FILES:

B. MANUSCRIPT ORGANIZATION AND FORMATTING:

Sincerely,

May 10, 2023

RE: Life Science Alliance Manuscript #LSA-2022-01813RRR

Prof. Eva Böttcher-Friebertshäuser
Philipps-Universität Marburg
Institute of Virology
Hans-Meerwein-Str. 2
Marburg 35043
Germany

Dear Dr. Böttcher-Friebertshäuser,

Thank you for submitting your Research Article entitled "Expression of Tmprss2 is upregulated by bacterial flagellin, LPS and Pam3Cys in human airway cells". It is a pleasure to let you know that your manuscript is now accepted for publication in Life Science Alliance. Congratulations on this interesting work.

DISTRIBUTION OF MATERIALS:

Again, congratulations on a very nice paper. I hope you found the review process to be constructive and are pleased with how the manuscript was handled editorially. We look forward to future exciting submissions from your lab.

Sincerely,
